# Medical costs for patients with rheumatoid arthritis who have comorbid diabetes mellitus

**Eiichi Tanaka‚[1]\*, Eisuke Inoue[1,2], Ryoko Sakai[1,3], Katsuhiko Iwasaki[4,5], Ayako Shoji[4,5,6], Masayoshi Harigai[1¤]**

1 Department of Rheumatology, Tokyo Women's Medical University School of Medicine, Shinjuku, Tokyo, Japan, 2 Showa University Research Administration Center, Showa University, Shinagawa, Tokyo, Japan, 3 Department of Public Health and Epidemiology, Meiji Pharmaceutical University, Kiyose-City, Tokyo, Japan, 4 Division of Science and Analytics, Medilead Inc, Shinjuku, Tokyo, Japan, 5 Healthcare Consulting, Inc, Chiyoda, Tokyo, Japan, 6 Department of Health Economics and Outcomes Research, Graduate School of Pharmaceutical Sciences, The University of Tokyo, Bunkyo, Tokyo, Japan

¤ Current address: Masayoshi Harigai Sanno Medical Centre, 8-5-35 Akasaka, Tokyo, 107−0052, Japan
\* e-tanaka@twmu.ac.jp

## Abstract

### Objectives

To evaluate medical costs and resource use in patients with rheumatoid arthritis (RA) with and without diabetes mellitus (DM).

### Methods

Data were obtained from the Japan Medical Data Center (JMDC) claims database. The baseline period comprised 6 months before the index date (first prescription date post-RA diagnosis) while the 12-month duration post-index date was the follow-up period. Patients with RA and DM prescribed an antidiabetic drug in the baseline period constituted the DM group, while patients with RA without DM or an antidiabetic prescription in the baseline and follow-up periods belonged to the non-DM group. Patients were matched by sex, age, Charlson Comorbidity Index (CCI), months from the first RA codes, and medications. The primary endpoint was total medical costs per patient in the follow-up period. The secondary endpoints were costs for drugs, treatments, and materials, their sub-categories, and proportions of patients using the sub-categories.

### Results

One hundred and sixty-one DM group patients and 2,974 non-DM group patients were eligible for inclusion, and 109 patients were matched from each group. The median values of age and CCI were 59 years and 2.0 in both groups. After excluding DM-specific costs, both total medical costs and drug costs were significantly higher in the DM group compared with the non-DM group (total medical costs: DM/

**Data availability statement:** Data cannot be shared publicly because of ethical restrictions involving patient information. Only the minimum data set can be shared with qualified researchers who meet the criteria upon request via email e-tanaka@twmu.ac.jp. We are not able to share a de-identified data set because the data that we used owned by JMDC Inc. We provide the non-author contact information, JMDC Inc. (https://www.jmdc.co.jp/inquiry/) from which the data also can be provided.

**Funding:** We would like to update the Funding Statement section as follows: Funding for this study was provided by Bristol-Myers Squibb K.K. The funder did not provide support in the form of salaries for any author. The funders had no role in study design, data collection and analysis, decision to publish, or preparation of the manuscript.

**Competing interests:** ET has received lecture fees or consulting fees from AbbVie Japan GK, Asahi Kasei Corp., Astellas Pharma Inc., Ayumi Pharmaceutical Co., Boehringer Ingelheim Japan, Inc., Bristol Myers Squibb Co., Ltd., Chugai Pharmaceutical Co., Ltd., Daiichi-Sankyo, Inc., Eisai Co., Ltd., Eli Lilly Japan K.K., Gilead Sciences, Inc., Pfizer Japan Inc, Nichi-Iko Pharmaceutical Co., Ltd., Taisho Pharmaceutical Co., Ltd, Takeda Pharmaceutical Co., Ltd, Mitsubishi Tanabe Pharma Co., UCB Japan Co. Ltd. and Viatris Inc. ET has received research funding from Pfizer Inc. and UCB Japan Co. Ltd. EI received lecture fees from Eisai Co., Ltd. and Chugai Pharmaceutical Co., Ltd. RS has nothing to declare. KI was an employee of Medilead, Inc., which was commissioned to perform this study analysis by Tokyo Women's Medical University. During the study, KI was affiliated with Medilead, Inc., but is currently employed by Healthcare Consulting Inc. AS was an employee of Medilead, Inc., which was commissioned to perform this study analysis by Tokyo Women's Medical University. During the study, KI AS was affiliated with Medilead, Inc., but is currently employed by Healthcare Consulting Inc and the University of Tokyo. MH has received research grants from AbbVie Japan GK, Asahi Kasei Corp., Ayumi Pharmaceutical Co., Boehringer Ingelheim Japan, Inc., Bristol Myers Squibb Co., Ltd., Chugai Pharmaceutical Co., Eisai Co., Ltd., Eli Lilly Japan K.K., Kaken

non-DM: 5,163 USD/ 3,782 USD, $P < 0.05$; drug costs: DM/ non-DM: 2,242 USD/ 1,066 USD, $P < 0.05$) because of a higher proportion of biological disease-modifying antirheumatic drug (DMARD) users in the DM group (DM/ non-DM: 14.7%/ 5.5%). Treatment costs and material costs did not differ between the two groups.

## Conclusions

Medical costs for RA were higher in the DM group than in the non-DM group because of a higher proportion of biological DMARD users.

## Introduction

Rheumatoid arthritis (RA) is a chronic autoimmune disease characterized by synovitis and systemic inflammation. Chronic inflammation in RA is suggested to affect respiratory, cardiac, neurologic, and hematologic systems, among others [1]. Patients with RA may experience various comorbidities [2], among which much research has focused on cardiovascular diseases because of their prevalence and clinical relevance in patients with RA [3–7]. Comorbid diseases require the additional use of health care services not only for the treatment of comorbidities but also for the treatment of RA itself [2]. The rates of discontinuation of RA treatments due to adverse events and failure to achieve remission are high, and both were reported to be positively associated with the number and severity of comorbidities at baseline [8]. Comorbidities can also significantly increase the complexity of diagnosing RA and deciding on treatment, especially in case of atypical or silent signs and symptoms of comorbidities [2].

Diabetes mellitus (DM) is known to increase the risk of cardiovascular death, and its incidence is suggested to be higher in people with RA than in people without RA. A review found that the pathology of DM is likely associated with a key inflammatory process in the induction and progression of RA [9]. The increased occurrence of type 2 DM in the general populations of high-income countries is also contributing to the higher proportion of comorbid type 2 DM in RA. Comorbid DM is likely to affect treatment outcomes. Patients who have both RA and DM showed significantly higher disease activity after adjusting for age, sex, smoking status, other comorbidities, RA disease duration, and prescribed drugs [10].

Therefore, comorbid DM is expected to increase medical costs and the use of medical resources in patients with RA, but few previous studies have focused on this aspect. To confirm the impact of comorbid DM on the economic burden of treating patients with RA, this study aimed to compare the cost and amount of health care resources used with and without comorbid DM by analyzing data from a large-scale health insurance claims database in Japan.

## Materials and methods

### Study design

This was a retrospective cohort study that evaluated medical costs and use of health care resources in patients with RA with and without comorbid DM. The protocol

Pharmaceutical Co., Ltd., Mitsubishi Tanabe Pharma Co., Mochida Pharmaceutical Co., Ltd., Nippon Kayaku Co., Ltd., Pfizer Japan Inc., Taisho Pharmaceutical Co., Ltd., Teijin Pharma Ltd., UCB Japan Co., Ltd., and Viatris Japan. MH has received speaker's fee from AbbVie Japan GK, Asahi Kasei Corp., Astra Zeneca K. K., Ayumi Pharmaceutical Co., Boehringer Ingelheim Japan, Inc., Bristol Myers Squibb Co., Ltd., Chugai Pharmaceutical Co., Ltd., Eisai Co., Ltd., Eli Lilly Japan K.K., GlaxoSmithKline K.K., Gilead Sciences Inc., Janssen Pharmaceutical K.K., Mitsubishi Tanabe Pharma Co., Mochida Pharmaceutical Co., Ltd., Ono Pharmaceutical Co., Ltd., Pfizer Japan Inc., Taisho Pharmaceutical Co., Ltd., and Teijin Pharma Ltd. MH is a consultant for AbbVie, Boehringer-ingelheim, Bristol Myers Squibb Co., and Teijin Pharma. This does not alter our adherence to PLOS ONE policies on sharing data and materials.

including this study was approved by the Ethics Committee of Tokyo Women's Medical University on November 21, 2018 (Ethical Approval Number: 4991). Informed consent was not applicable for this study, based on the ethical guidelines for medical and health research involving human participants issued by the Ministry of Health, Labor and Welfare and Ministry of Education, Culture, Sports, Science and Technology of Japan (revised on February 28, 2017). The study period was from March 9, 2020 to February 26, 2024.

## Data source

We obtained data for this study from a large-scale Japanese health insurance claims database administered by the Japan Medical Data Center (JMDC Inc.) [11]. This database contains the accumulated health insurance claims data of Japanese workers employed at medium- to large-scale companies and the dependents of those employees. The data have been collected from health insurance companies since 2005, and at the end of 2018 the database contained data from 5.6 million Japanese individuals. The database contains anonymized information, such as patient characteristics (e.g. age, sex), claims data for outpatient, inpatient, and dispensing services, and clinical diagnoses coded by the International Classification of Diseases 10th revision (ICD-10). The database allows patients to be continuously followed, even if they were transferred to another hospital or visited multiple medical institutions, as long as the insurance system to which they belong does not change [11].

## Study population

All available medical and prescription records of eligible patients were extracted from the JMDC database for the study period extending from January 1, 2005, to April 30, 2019, as shown in the flowchart of patient identification (Fig 1). Patients with a confirmed diagnosis of RA were identified by the relevant ICD-10 code (shown in Supplementary S1 Table). Patients were included if they had at least one prescription of methotrexate (MTX), including biological disease-modifying antirheumatic drugs (bDMARDs), or targeted synthetic DMARDs (tsDMARDs) (shown in Supplementary S2 Table) after the first diagnosis of RA. The first prescription date after the first diagnosis of RA was defined as the index date. We included patients with an index date in the period from January 1, 2012, to December 31, 2017, and available data from 6 months before to 12 months after the index date; the 6 months period before the index date was defined as the baseline period and the 12 months duration after the index date was denoted as the follow-up period. Patients with at least one confirmed diagnosis of any of the diseases shown in Supplementary S1 Table in the baseline or the follow-up period were excluded, as were patients with at least one prescription of any DMARDs in the baseline period.

Patients with RA and both a confirmed diagnosis of DM (shown in S1 Table) and at least one prescription of any antidiabetic drug (shown in S3 Table) in the baseline period were defined as the DM group, and patients with no confirmed diagnosis of DM and no prescription of any antidiabetic drug in the baseline and follow-up periods were defined as the non-DM group (Fig 1). We then created a matched sample of patients by matching patients from each group at a 1:1 ratio on the basis of sex, age,

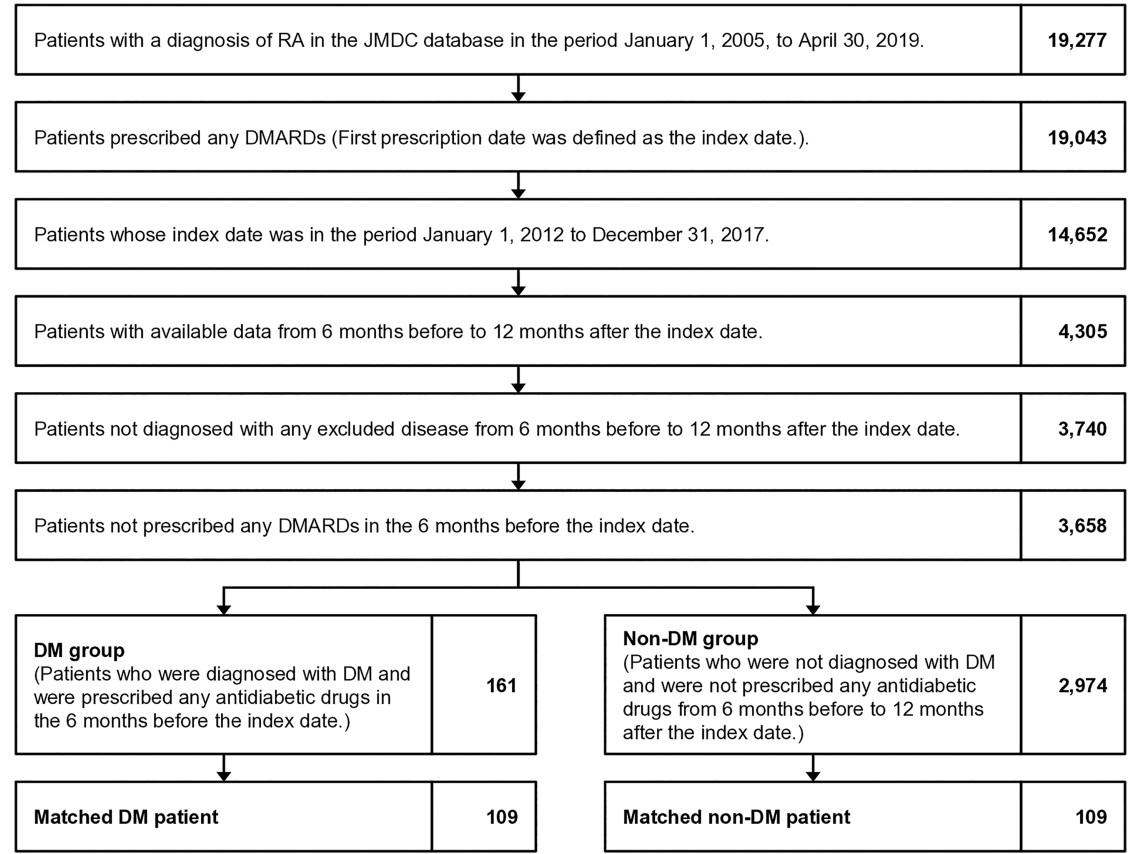

**Fig 1. Flowchart of patient identification.** DM, diabetes mellitus; DMARD, disease-modifying antirheumatic drug; JMDC, Japan Medical Data Center; RA, rheumatoid arthritis.

Charlson Comorbidity Index (CCI) [12], disease duration of RA (defined by the number of days from the first ICD-10 codes of RA to the index date), type of DMARDs (conventional synthetic DMARDs [csDMARDs], bDMARDs, or tsDMARDs) prescribed at the index date, corticosteroids (CSs) prescribed in the baseline period (shown in S2 Table), and use of anti-hypertensive drugs and anti-dyslipidemic drugs in the baseline period (shown in S3 Table). We used a 1:1 ratio to include the largest possible number of patients in the DM group; the number of matched patients in the DM group decreased when the number of patients in the non-DM group increased. In the matching process, we allowed differences of 5 years for age and 6 months for disease duration. When calculating the CCI, we excluded DM. In addition, human immunodeficiency virus infection and acquired immunodeficiency syndrome were also not considered because all information associated with their diagnosis and treatment had been removed from the database owing to the privacy policy of JMDC.

## Endpoints

The primary endpoint was the total medical cost per patient in the 12-month follow-up period. Medical costs in Japanese Yen (JPY) were converted into US Dollars (USD) (1 USD equaled 105 JPY in 2020, the exchange rate at the time of the analyses).

The secondary endpoints were medical costs in three categories: drug costs, treatment costs, and material costs. Drug costs included costs for DMARDs, CSs, analgesics, and antidiabetics (shown in S2 and S3 Tables). To assess treatment and material costs, we considered all costs for medical care during the follow-up period (shown in S4 and S5 Tables), with

some exceptions, such as cataract operations and childbirth-related medical care (shown in S4 and S6 Tables). We evaluated the medical costs with and without costs for DM-specific medical care. We considered some costs to be DM-specific, namely, costs for antidiabetic drugs and for the following items in three sub-categories of treatment costs: management of DM in the sub-category medical management; blood glucose self-monitoring and home self-injection in the sub-category home health care; and assessment of hemoglobin A1c (HbA1c) and glucose levels in the sub-category examination. None of the material costs were considered to be DM-specific.

Another secondary endpoint was the use of medical resources in the follow-up period for all reasons except DM-specific care; these were also categorized into drug, treatment, and material costs. The types of medical care considered in the secondary endpoints were similar to those for the primary endpoint. For the use of drugs, we assessed the number and proportion of patients who used each eligible drug in the follow-up period, while for use of treatment resources we assessed the number and proportion of patients who used each resource and the number of days the resource was used in the follow-up period.

### Statistical analysis

For medical costs, days of resource use, age, CCI, and disease duration, we calculated means (± standard deviation, SD) or medians (interquartile range, IQR) and performed Wilcoxon signed rank tests to assess differences in each variable between matched DM and non-DM groups. For use of drug and treatment resources, we calculated the number and proportion of patients and performed McNemar's test to assess group differences. The results were considered to be statistically significant when the *p*-value was less than 0.05. We performed all comparisons with two-sided tests and conducted all statistical analyses with Statistical Analysis Software R (version 3.6.0).

## Results

### Patient characteristics

A total of 161 patients were eligible for inclusion in the DM group and 2,974 were eligible for inclusion in the non-DM group (Fig 1). We found differences between the groups regarding sex, age, CCI, and treatment with antihypertensive and anti-dyslipidemic drugs in the baseline period (Table 1). After matching patients, we were able to include 109 from each group (Fig 1); no differences in patient backgrounds were present between the two matched groups (Table 1).

### Medical costs

The total medical costs and their three categories (drugs, treatments, and materials) are listed in Table 2. Regardless of the DM-specific costs, the total medical costs and the costs for drugs were significantly higher in the DM group compared with the non-DM group (Table 2).

The costs for the various sub-categories of drugs are shown in Table 3. In both groups, the costs for bDMARDs constituted a high proportion of the total drug costs. In the DM group, the costs of bDMARDs accounted for a high proportion of the total drug costs, and the costs of tumor necrosis factor inhibitors (TNFi) were particularly high (Table 3). The costs for all bDMARDs, all analgesics, and non-steroidal anti-inflammatory drugs (NSAIDs) were significantly higher in the DM group compared with the non-DM group (Table 3).

The costs for the various subcategories of treatment, excluding those for DM-specific treatment, are shown in Table 4. In both groups, examination costs accounted for the highest proportion of the total treatment costs (Table 4), but the examination costs in the DM group were significantly higher than in the non-DM group (Table 4). If the costs of DM-specific medical care were included, the mean (± SD) treatment costs in the various subcategories in the DM vs non-DM groups were as follows: medical management, 231 (201) USD vs 213 (201) USD (*P*=0.354); home health care, 338 (642) USD vs 10 (59) USD (*P*<0.001); examination, 1,225 (581) USD vs 946 (566) USD (*P*<0.001). The costs for the other treatment sub-categories that were not associated with DM-specific medical care were the same as shown in Table 4.

**Table 1. Patient characteristics.**

| Characteristic | | Before matching | | | After matching | | |
|---|---|---|---|---|---|---|---|
| | | DM (N = 161) | Non-DM (N = 2,974) | P-value | DM (N = 109) | Non-DM (N = 109) | P-value |
| Sex, n (%) | Female | 78 (48.4) | 2,184 (73.4) | < 0.001 | 55 (50.5) | 55 (50.5) | 1.000 |
| DMARDs at index date, n (%) | csDMARDs | 155 (96.3) | 2,894 (97.3) | 0.591 | 109 (100.0) | 109 (100.0) | 1.000 |
| | bDMARDs | 8 (5.0) | 130 (4.4) | 0.871 | 0 (0.0) | 0 (0.0) | 1.000 |
| | tsDMARDs | 0 (0.0) | 0 (0.0) | NA | 0 (0.0) | 0 (0.0) | 1.000 |
| Medication in baseline period, n (%) | CSs | 85 (52.8) | 1,336 (44.9) | 0.061 | 53 (48.6) | 53 (48.6) | 1.000 |
| | Antihypertensive drugs | 82 (50.9) | 320 (10.8) | < 0.001 | 49 (45.0) | 49 (45.0) | 1.000 |
| | Antidyslipidemic drugs | 77 (47.8) | 301 (10.1) | < 0.001 | 48 (44.0) | 48 (44.0) | 1.000 |
| Age, median (IQR), years old | | 59.0 (53.0 - 63.0) | 49.0 (42.0 - 56.0) | < 0.001 | 59.0 (54.0 - 63.0) | 59.0 (54.0 - 62.0) | 0.740 |
| CCI, median (IQR) | | 2.0 (1.0 - 3.0) | 1.0 (1.0 - 2.0) | < 0.001 | 2.0 (1.0 - 2.0) | 2.0 (1.0 - 2.0) | NA |
| Disease duration, median (IQR), months | | 1.2 (0.5 - 5.8) | 1.0 (0.6 - 6.9) | 0.630 | 0.9 (0.5 - 2.9) | 0.9 (0.5 - 2.1) | 0.340 |

bDMARD, biological disease-modifying antirheumatic drug; CCI, Charlson comorbidity index; CS, corticosteroid; csDMARD, conventional synthetic disease-modifying antirheumatic drug; DM, diabetes mellitus; IQR, interquartile range; NA, not applicable; tsDMARD, targeted synthetic disease-modifying antirheumatic drug.

P-value before matching: Chi-square test for sex, DMARDs at index date, and medication in the baseline period; Wilcoxon rank-sum test for age, CCI, and disease duration.

P-value after matching: McNemar's test for sex, DMARDs at index date, and medication in the baseline period; Wilcoxon signed-rank test for age, CCI, and disease duration.

**Table 2. Annual medical costs per patient.**

| Type of medical cost | | Medical costs per patient, mean (SD), USD/year | | |
|---|---|---|---|---|
| | | DM (N = 109) | Non-DM (N = 109) | P-value |
| Including DM costs | Total | 6,347 (5,234) | 3,810 (4,050) | < 0.001 |
| | Drug costs | 3,032 (3,877) | 1,066 (1,712) | < 0.001 |
| | Treatment costs | 3,227 (2,595) | 2,567 (2,217) | 0.001 |
| | Material costs | 88 (576) | 177 (1,006) | 0.081 |
| Excluding DM costs | Total | 5,163 (5,017) | 3,782 (4,018) | < 0.001 |
| | Drug costs | 2,242 (3,816) | 1,066 (1,712) | 0.008 |
| | Treatment costs | 2,833 (2,451) | 2,539 (2,191) | 0.076 |
| | Material costs | 88 (576) | 177 (1,006) | 0.081 |

DM, diabetes mellitus; SD, standard deviation.

1 USD = 105 JPY in 2020.

P-value: Wilcoxon signed rank test.

## Medical resource use

The number and proportion of patients who used the various types of RA-related drugs in the follow-up period is shown in Table 5. The proportion of MTX users was similar between the two groups, but that of bDMARDs users was significantly higher in the DM group compared with the non-DM group (Table 5).

The numbers and proportions of patients who received each type of treatment in the follow-up period, excluding DM-specific treatment, are shown in Table 6. No difference was found between the two patient groups in any sub-category (Table 6).

**Table 3. Annual drug costs per patient.**

| Type of drug | | Drug cost per patient, mean (SD), USD/year | | |
|---|---|---|---|---|
| | | DM (N = 109) | Non-DM (N = 109) | *P*-value |
| csDMARDs | Total | 550 (592) | 474 (396) | 0.857 |
| | MTX | 321 (227) | 303 (223) | 0.449 |
| | Others | 229 (586) | 171 (379) | 0.700 |
| bDMARDs | Total | 1,376 (3,619) | 348 (1,546) | 0.011 |
| | TNFi | 845 (2,944) | 284 (1,478) | 0.088 |
| | IL6i | 296 (1,351) | 65 (494) | 0.063 |
| | T-cell | 235 (1,342) | 0 (0) | 0.068 |
| tsDMARDs | | 0 (0) | 0 (0) | NA |
| CSs | | 28 (46) | 32 (59) | 0.545 |
| Analgesics | Total | 288 (317) | 212 (336) | 0.007 |
| | AAP | 2 (19) | 4 (19) | 0.315 |
| | AAP/Opioids | 20 (98) | 19 (105) | 0.605 |
| | NSAIDs | 229 (267) | 167 (261) | 0.012 |
| | Opioids | 0 (0) | 2 (16) | 0.068 |
| | Others | 36 (107) | 20 (74) | 0.098 |
| DM | | 790 (602) | 0 (0) | < 0.001 |

AAP, acetaminophen; bDMARD, biological disease-modifying antirheumatic drug; CS, corticosteroid; csDMARD, conventional synthetic disease-modifying antirheumatic drug; DM, diabetes mellitus; IL6i, interleukin-6 inhibitor; MTX, methotrexate; NA, not applicable; NSAID, nonsteroidal anti-inflammatory drug; SD, standard deviation; T-cell, selective T-cell co-stimulation modulator; TNFi, tumor necrosis factor inhibitor; tsDMARD, targeted synthetic disease-modifying antirheumatic drug.

1 USD = 105 JPY in 2020.

*P*-value: Wilcoxon signed rank test.

The number of days when patients used treatment resources excluding DM-specific medical care during the follow-up period are shown in Table 7. No difference was found between the two patient groups for any type of treatment resources (Table 7).

## Discussion

The present study found that, after matching patient characteristics by sex, age, CCI, disease duration of RA, and medication, medical costs in patients with RA were higher in the DM group than in the non-DM group. To our knowledge, this is the first study that has demonstrated the additional medical costs in patients with RA and DM.

Some of the differences between the two patient groups were attributable to treatments for DM, but even after DM-specific medical costs were excluded, drug costs in the DM group were approximately twice as high as those in the non-DM group because of significantly higher expenses for bDMARDs. A higher proportion of patients in the DM group were prescribed bDMARDs, and the unit price of these drugs is much higher than that of other drugs. In general, patients with RA are known to initiate treatment with a bDMARD if they show insufficient improvement with csDMARDs including MTX or if they cannot use or tolerate csDMARDs such as MTX. Our results suggest that failure to control disease activity with an initial therapy may be more common in the DM than in the non-DM group because we found no difference between the two patient groups in the proportion of patients using MTX in the follow-up period. Some previous studies have suggested that comorbid DM is likely associated with higher disease activity and more frequent failure of csDMARDs treatment [10,13]. One study assessed the relationship between cardiovascular comorbidities including DM and disease activity in patients with RA by multivariate analyses adjusted for major confounders and found a positive association of DM with high

**Table 4. Annual treatment costs per patient excluding DM-specific medical practices.**

| Type of treatment | Treatment costs per patient, mean (SD), USD/year | | |
|---|---|---|---|
| | DM (N = 109) | Non-DM (N = 109) | *P*-value |
| Outpatient | 355 (219) | 332 (202) | 0.452 |
| Hospitalization | 335 (1,144) | 151 (600) | 0.148 |
| Medical management | 213 (181) | 213 (201) | 0.707 |
| Home health care | 32 (226) | 0 (0) | 0.109 |
| Examination | 1,155 (560) | 927 (556) | 0.001 |
| Imaging | 205 (253) | 259 (310) | 0.141 |
| Medication | 194 (99) | 177 (96) | 0.197 |
| Injection | 51 (127) | 24 (57) | 0.058 |
| Rehabilitation | 47 (168) | 71 (237) | 0.465 |
| Psychiatric specialty therapy | 8 (49) | 32 (180) | 0.208 |
| Procedure | 32 (70) | 25 (60) | 0.482 |
| Surgery | 152 (808) | 229 (948) | 0.200 |
| Anesthesia | 24 (137) | 53 (214) | 0.340 |
| Radiation therapy | 0 (0) | 0 (0) | NA |
| Pathological diagnosis | 30 (89) | 45 (116) | 0.231 |
| Others | 1 (4) | 0 (5) | 0.208 |

DM, diabetes mellitus; NA, not applicable; SD, standard deviation.

1 USD = 105 JPY in 2020.

*P*-value: Wilcoxon signed rank test.

**Table 5. Number and proportion of patients who used drugs.**

| Type of drug | | Drug use, n (%) | | |
|---|---|---|---|---|
| | | DM (N = 109) | Non-DM (N = 109) | *P*-value |
| csDMARDs | Total | 109 (100.0) | 109 (100.0) | 1.000 |
| | MTX | 101 (92.7) | 102 (93.6) | 1.000 |
| | Others | 46 (42.2) | 51 (46.8) | 0.583 |
| bDMARDs | Total | 16 (14.7) | 6 (5.5) | 0.041 |
| | TNFi | 11 (10.1) | 4 (3.7) | 0.118 |
| | IL6i | 6 (5.5) | 2 (1.8) | 0.219 |
| | T-cell | 4 (3.7) | 0 (0.0) | 0.125 |
| tsDMARDs | | 0 (0.0) | 0 (0.0) | 1.000 |
| CSs | | 65 (59.6) | 62 (56.9) | 0.711 |
| Analgesics | Total | 103 (94.5) | 96 (88.1) | 0.167 |
| | AAP | 24 (22.0) | 23 (21.1) | 1.000 |
| | AAP/Opioids | 10 (9.2) | 6 (5.5) | 0.454 |
| | NSAIDs | 102 (93.6) | 93 (85.3) | 0.093 |
| | Opioids | 0 (0.0) | 4 (3.7) | 0.125 |
| | Others | 25 (22.9) | 17 (15.6) | 0.185 |

AAP, acetaminophen; bDMARD, biological disease-modifying antirheumatic drug; CSs, corticosteroids; csDMARD, conventional synthetic disease-modifying antirheumatic drug; DM, diabetes mellitus; IL6i, interleukin-6 inhibitor; MTX, methotrexate; NSAID, non-steroidal anti-inflammatory drug; T-cell, selective T-cell costimulation modulator; TNFi, tumor necrosis factor inhibitor; tsDMARD, targeted synthetic disease-modifying antirheumatic drug

*P*-value: derived from McNemar's test.

**Table 6. Number and proportion of patients with treatment resource use excluding DM-specific medical care.**

| Type of treatment | Treatment resource use, n (%) | | |
|---|---|---|---|
| | DM (N = 109) | Non-DM (N = 109) | *P*-value |
| Outpatient | 109 (100.0) | 109 (100.0) | 1.000 |
| Hospitalization | 25 (22.9) | 18 (16.5) | 0.281 |
| Medical management | 102 (93.6) | 98 (89.9) | 0.481 |
| Home health care | 3 (2.8) | 0 (0.0) | 0.250 |
| Examination | 109 (100.0) | 108 (99.1) | 1.000 |
| Imaging | 90 (82.6) | 98 (89.9) | 0.134 |
| Medication | 109 (100.0) | 109 (100.0) | 1.000 |
| Injection | 56 (51.4) | 53 (48.6) | 0.780 |
| Rehabilitation | 16 (14.7) | 17 (15.6) | 1.000 |
| Psychiatric specialty therapy | 4 (3.7) | 5 (4.6) | 1.000 |
| Procedure | 52 (47.7) | 52 (47.7) | 1.000 |
| Surgery | 16 (14.7) | 21 (19.3) | 0.442 |
| Anesthesia | 13 (11.9) | 15 (13.8) | 0.839 |
| Radiation therapy | 20 (18.3) | 32 (29.4) | 0.088 |
| Pathological diagnosis | 6 (5.5) | 2 (1.8) | 0.289 |
| Others | 109 (100.0) | 109 (100.0) | 1.000 |

DM, diabetes mellitus.

*P*-value: McNemar test.

**Table 7. Number of days when patients used various types of treatment resources excluding DM-specific medical care.**

| Type of treatment | Treatment resource use, mean (SD), days/year | | |
|---|---|---|---|
| | DM (N = 109) | Non-DM (N = 109) | *P*-value |
| Outpatient | 31.7 (19.6) | 30.0 (21.0) | 0.300 |
| Hospitalization | 2.6 (8.2) | 1.9 (8.1) | 0.127 |
| Imaging | 2.9 (3.1) | 3.6 (3.9) | 0.068 |
| Rehabilitation | 2.6 (8.9) | 2.8 (9.4) | 0.918 |
| Procedure | 5.4 (13.7) | 5.2 (16.0) | 0.951 |
| Surgery | 0.2 (0.5) | 0.2 (0.6) | 0.530 |

DM, diabetes mellitus; SD, standard deviation.

*P*-value: Wilcoxon signed rank test.

disease activity [10]. This association was similarly observed in all activity measures investigated. Another study showed that more than 95% of patients with RA and DM used glucocorticoids and suggested that patients with RA and DM were likely to have more severe inflammation [13]. In contrast, many studies have suggested that TNFis decrease insulin resistance and thus decrease the risk of developing DM [14]. Another study suggested that a T-cell co-stimulation modulator was associated with the reduction in DM-related medical costs [15]. An improved understanding of the molecular mechanisms behind the accelerated inflammation observed in patients with RA and DM may lead to a better treatment strategy.

Our results showed that in the follow-up period, the proportion of NSAID users was comparable in the two patient groups but the costs for NSAIDs were significantly higher in the DM group. Although we cannot assert that NSAIDs were prescribed for the purpose of relieving pain associated with RA, these results suggest that this patient cohort may have had more severe disease inflammation than the non-DM group and therefore required larger amounts of NSAIDs.

Although surgical procedures can also be an outcome of disease inflammation and progression, the proportion of patients who underwent an operation did not differ between the two groups; however, we hypothesize that if the study population was followed for a longer period of time, we may find a higher incidence rate for surgery in the DM group.

Treatment costs were also significantly higher in the DM group compared with the non-DM group. In particular, costs for examinations were higher in the DM group, although the proportion of treatment users did not significantly differ between the two groups. Comorbidity with DM requires additional examinations not only for glycemic control but also for risks of DM-specific vascular disorders; these examinations include blood tests and heart, kidney, and liver function tests. Frequent examinations, including expensive tests, increased treatment costs in the DM group. Although imaging procedures are also likely to be required in patients with DM to test for micro- or macroangiopathy, imaging costs were not significantly different between the two groups. However, we adjusted for demographic and clinical characteristics, and hence patients with advanced DM who are more likely to have such vascular disorders, may have been excluded from the study population. Comorbidity with DM is considered to increase the risk of infection, but the proportion of patients who were hospitalized and the days spent in hospital did not show any differences between the two patient groups in the 12-month follow-up period.

## Limitations

The present study has some limitations. First, the patients with RA we analyzed included those who had at least one prescription of MTX, bDMARDs, or tsDMARDs and excluded those who used only non-MTX csDMARDs namely, bucillamine or salazosulfapyridine. Second, we chose a 1:1 matching ratio between the two patient groups to maximize the number of patients in the DM group. Therefore, approximately 95% of patients in the non-DM group were excluded from the analyses. After a 1:2 and 1:3 ratio of patient matching, the resulting numbers of included patients in the DM group were 80 and 62, respectively. We confirmed that the ratio of patient matching had no impact on our results (see S7 Table). Third, the claims database we used does not contain any information on the severity of RA, such as the Disease Activity Score, Clinical Disease Activity Index, Simplified Disease Activity Index, and Health Assessment Questionnaire score, so disease severity may have differed in the two patient populations even though the matching process considered prescribed drugs at baseline; however, we believe that this aspect had only limited impact on our conclusions because the proportion of users of csDMARDs and CSs was comparable in the follow-up period. Fourth, although more than 70% of patients with RA are women, only about 50% of our patient-matched groups were women. In addition, because only corporate employees and their families were registered in the JMDC database, the proportion of elderly people aged 65 and over, who are the non-working population, was relatively small. Therefore, our findings may not represent the general population with RA. Lastly, we used ICD-10 codes to identify patients with RA, but the validity of these codes has not been confirmed in the JMDC claims database.

## Conclusions

Patients with RA who have comorbid DM were more likely to use relatively expensive bDMARDs, resulting in higher medical costs than those patients with RA who do not have comorbid DM.

## Supporting information

**S1 Table. ICD-10 codes.**
(PDF)

**S2 Table. Items included in drug costs for treatment of RA.**
(PDF)

**S3 Table. Items included in drug costs for treatment of comorbidities.**
(PDF)

**S4 Table. Items included in treatment costs.**
(PDF)

**S5 Table. Items included in material costs.**
(PDF)

**S6 Table. List of excluded medical cares.**
(PDF)

**S7 Table. Sensitivity analysis of total medical cost by matching ratio of non-DM group against DM group.**
(PDF)

## Acknowledgments

We would like to thank Yamada Translation Bureau, Inc. (https://www.ytrans.com/home.html) for English language editing.

## Author contributions

Conceptualization: Eiichi Tanaka.

Data curation: Katsuhiko Iwasaki, Ayako Shoji.

Formal analysis: Eisuke Inoue, Katsuhiko Iwasaki, Ayako Shoji.

Funding acquisition: Masayoshi Harigai.

Investigation: Eiichi Tanaka, Ryoko Sakai.

Methodology: Eisuke Inoue, Ryoko Sakai, Katsuhiko Iwasaki, Ayako Shoji.

Project administration: Eiichi Tanaka.

Resources: Eiichi Tanaka.

Software: Katsuhiko Iwasaki, Ayako Shoji.

Supervision: Masayoshi Harigai.

Writing – original draft: Eiichi Tanaka, Ayako Shoji.

Writing – review & editing: Eisuke Inoue, Ryoko Sakai, Katsuhiko Iwasaki, Masayoshi Harigai.

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
