## [Decision Letter · Decision Letter 0]

Dear Dr. Tanaka,

Thank you for submitting your manuscript to PLOS ONE. After careful consideration, we feel that it has merit but does not fully meet PLOS ONE’s publication criteria as it currently stands. Therefore, we invite you to submit a revised version of the manuscript that addresses the points raised during the review process.

We look forward to receiving your revised manuscript.

Kind regards,

Luca Navarini

Academic Editor

PLOS ONE

**Journal Requirements:**

1. When submitting your revision, we need you to address these additional requirements. Please ensure that your manuscript meets PLOS ONE's style requirements, including those for file naming. The PLOS ONE style templates can be found at https://journals.plos.org/plosone/s/file?id=wjVg/PLOSOne_formatting_sample_main_body.pdf and https://journals.plos.org/plosone/s/file?id=ba62/PLOSOne_formatting_sample_title_authors_affiliations.pdf 2. Thank you for stating the following financial disclosure: Funding for this study was provided by Bristol-Myers Squibb K.K. Please state what role the funders took in the study.  If the funders had no role, please state: "The funders had no role in study design, data collection and analysis, decision to publish, or preparation of the manuscript." If this statement is not correct you must amend it as needed. Please include this amended Role of Funder statement in your cover letter; we will change the online submission form on your behalf. 3. Thank you for stating the following in the Competing Interests section:  ET has received lecture fees or consulting fees from AbbVie Japan GK, Asahi Kasei Corp., Astellas Pharma Inc., Ayumi Pharmaceutical Co., Boehringer Ingelheim Japan, Inc., Bristol Myers Squibb Co., Ltd., Chugai Pharmaceutical Co., Ltd., Daiichi-Sankyo, Inc., Eisai Co., Ltd., Eli Lilly Japan K.K., Gilead Sciences, Inc., Pfizer Japan Inc, Nichi-Iko Pharmaceutical Co., Ltd., Taisho Pharmaceutical Co., Ltd, Takeda Pharmaceutical Co., Ltd, Mitsubishi Tanabe Pharma Co., UCB Japan Co. Ltd. and Viatris Inc. ET has received research funding from Pfizer Inc. and UCB Japan Co. Ltd. EI received lecture fees from Eisai Co., Ltd. and Chugai Pharmaceutical Co., Ltd.  RS has nothing to declare.  KI was an employee of Medilead, Inc., which was commissioned to perform this study analysis by Tokyo Women's Medical University. During the study, KI was affiliated with Medilead, Inc., but is currently employed by Healthcare Consulting Inc. AS was an employee of Medilead, Inc., which was commissioned to perform this study analysis by Tokyo Women's Medical University. During the study, KI was affiliated with Medilead, Inc., but is currently employed by Healthcare Consulting Inc and the University of Tokyo. MH has received research grants from AbbVie Japan GK, Asahi Kasei Corp., Ayumi Pharmaceutical Co., Boehringer Ingelheim Japan, Inc., Bristol Myers Squibb Co., Ltd., Chugai Pharmaceutical Co., Eisai Co., Ltd., Eli Lilly Japan K.K., Kaken Pharmaceutical Co., Ltd., Mitsubishi Tanabe Pharma Co., Mochida Pharmaceutical Co., Ltd., Nippon Kayaku Co., Ltd., Pfizer Japan Inc., Taisho Pharmaceutical Co., Ltd., Teijin Pharma Ltd., UCB Japan Co., Ltd., and Viatris Japan. MH has received speaker’s fee from AbbVie Japan GK, Asahi Kasei Corp., Astra Zeneca K. K., Ayumi Pharmaceutical Co., Boehringer Ingelheim Japan, Inc., Bristol Myers Squibb Co., Ltd., Chugai Pharmaceutical Co., Ltd., Eisai Co., Ltd., Eli Lilly Japan K.K., GlaxoSmithKline K.K., Gilead Sciences Inc., Janssen Pharmaceutical K.K., Mitsubishi Tanabe Pharma Co., Mochida Pharmaceutical Co., Ltd., Ono Pharmaceutical Co., Ltd., Pfizer Japan Inc., Taisho Pharmaceutical Co., Ltd., and Teijin Pharma Ltd. MH is a consultant for AbbVie, Boehringer-ingelheim, Bristol Myers Squibb Co., and Teijin Pharma. We note that one or more of the authors are employed by a commercial company.  a. Please provide an amended Funding Statement declaring this commercial affiliation, as well as a statement regarding the Role of Funders in your study. If the funding organization did not play a role in the study design, data collection and analysis, decision to publish, or preparation of the manuscript and only provided financial support in the form of authors' salaries and/or research materials, please review your statements relating to the author contributions, and ensure you have specifically and accurately indicated the role(s) that these authors had in your study. You can update author roles in the Author Contributions section of the online submission form. Please also include the following statement within your amended Funding Statement. “The funder provided support in the form of salaries for authors, but did not have any additional role in the study design, data collection and analysis, decision to publish, or preparation of the manuscript. The specific roles of these authors are articulated in the ‘author contributions’ section.”If your commercial affiliation did play a role in your study, please state and explain this role within your updated Funding Statement.  b. Please also provide an updated Competing Interests Statement declaring this commercial affiliation along with any other relevant declarations relating to employment, consultancy, patents, products in development, or marketed products, etc.   Within your Competing Interests Statement, please confirm that this commercial affiliation does not alter your adherence to all PLOS ONE policies on sharing data and materials by including the following statement: "This does not alter our adherence to  PLOS ONE policies on sharing data and materials.” (as detailed online in our guide for authors http://journals.plos.org/plosone/s/competing-interests) . If this adherence statement is not accurate and  there are restrictions on sharing of data and/or materials, please state these. Please note that we cannot proceed with consideration of your article until this information has been declared. Please include both an updated Funding Statement and Competing Interests Statement in your cover letter. We will change the online submission form on your behalf. 4. We note that you have indicated that there are restrictions to data sharing for this study. For studies involving human research participant data or other sensitive data, we encourage authors to share de-identified or anonymized data. However, when data cannot be publicly shared for ethical reasons, we allow authors to make their data sets available upon request. For information on unacceptable data access restrictions, please see http://journals.plos.org/plosone/s/data-availability#loc-unacceptable-data-access-restrictions.  Before we proceed with your manuscript, please address the following prompts: a) If there are ethical or legal restrictions on sharing a de-identified data set, please explain them in detail (e.g., data contain potentially identifying or sensitive patient information, data are owned by a third-party organization, etc.) and who has imposed them (e.g., a Research Ethics Committee or Institutional Review Board, etc.). Please also provide contact information for a data access committee, ethics committee, or other institutional body to which data requests may be sent. b) If there are no restrictions, please upload the minimal anonymized data set necessary to replicate your study findings to a stable, public repository and provide us with the relevant URLs, DOIs, or accession numbers. Please see http://www.bmj.com/content/340/bmj.c181.long for guidelines on how to de-identify and prepare clinical data for publication. For a list of recommended repositories, please see https://journals.plos.org/plosone/s/recommended-repositories. You also have the option of uploading the data as Supporting Information files, but we would recommend depositing data directly to a data repository if possible. Please update your Data Availability statement in the submission form accordingly. 5. In this instance it seems there may be acceptable restrictions in place that prevent the public sharing of your minimal data. However, in line with our goal of ensuring long-term data availability to all interested researchers, PLOS’ Data Policy states that authors cannot be the sole named individuals responsible for ensuring data access (http://journals.plos.org/plosone/s/data-availability#loc-acceptable-data-sharing-methods). Data requests to a non-author institutional point of contact, such as a data access or ethics committee, helps guarantee long term stability and availability of data. Providing interested researchers with a durable point of contact ensures data will be accessible even if an author changes email addresses, institutions, or becomes unavailable to answer requests. Before we proceed with your manuscript, please also provide non-author contact information (phone/email/hyperlink) for a data access committee, ethics committee, or other institutional body to which data requests may be sent. If no institutional body is available to respond to requests for your minimal data, please consider if there any institutional representatives who did not collaborate in the study, and are not listed as authors on the manuscript, who would be able to hold the data and respond to external requests for data access? If so, please provide their contact information (i.e., email address). Please also provide details on how you will ensure persistent or long-term data storage and availability. 6. When completing the data availability statement of the submission form, you indicated that you will make your data available on acceptance. We strongly recommend all authors decide on a data sharing plan before acceptance, as the process can be lengthy and hold up publication timelines. Please note that, though access restrictions are acceptable now, your entire data will need to be made freely accessible if your manuscript is accepted for publication. This policy applies to all data except where public deposition would breach compliance with the protocol approved by your research ethics board. If you are unable to adhere to our open data policy, please kindly revise your statement to explain your reasoning and we will seek the editor's input on an exemption. Please be assured that, once you have provided your new statement, the assessment of your exemption will not hold up the peer review process. 7. Please include a copy of Table S6 which you refer to in your text on page 486.

Reviewers' comments:

Reviewer's Responses to Questions

**Comments to the Author**

1. Is the manuscript technically sound, and do the data support the conclusions?

Reviewer #1: Yes

Reviewer #2: Yes

2. Has the statistical analysis been performed appropriately and rigorously?

Reviewer #1: Yes

Reviewer #2: Yes

3. Have the authors made all data underlying the findings in their manuscript fully available?

Reviewer #1: Yes

Reviewer #2: No

4. Is the manuscript presented in an intelligible fashion and written in standard English?

Reviewer #1: Yes

Reviewer #2: Yes

**Reviewer #1: ** The study is very interesting and the results obtained have been satisfactorily discussed. Further studies would be needed that take into account the limitations of the current study, as clarified by the Authors themselves.

**Reviewer #2:**  The study is well written and benefits from stringent inclusion criteria regarding ICD10 codes. Also, the limitations of the study have been thoroughly addressed in the last sections. The limitation to data availability has also been adequately justified. I would like to point out a few points that might need some additional explanation or revisiting:

- Lines 214-216: when assessing treatment costs you specified some exclusions, such as cataract operations. Would it be possible, if achievable, to provide a list of the excluded costs (maybe in a supplementary table) or, alternatively, declare the general criterion behind the choice of excluding some costs?

-Lines 365-366: "NSAIDs are commonly used in patients with RA, but patients often require higher doses to reduce inflammation". This affirmation, while being plausible according to clinical experience, seems to be lacking solid evidence in literature from comparative studies. In the absence of a citation that may back this statement, I would suggest to rephrase or delete the sentence, if not necessary for the narrative of your discussion.

Finally, the list of supplementary tables at the end of the document mentions a table S6: Sensitivity analysis of total medical cost by matching ratio of non-DM group against DM group, which apparently was not included in the downloadable xlsx file.

Overall, the article is well-structured and it will certainly provide useful insights of the topic after incorporating the suggested revisions.

**Do you want your identity to be public for this peer review?** For information about this choice, including consent withdrawal, please see our Privacy Policy

Reviewer #1: No

Reviewer #2: **Yes: ** Antonio Ciancio, MD

---

## [Author Response · Author response to Decision Letter 1]

27 May 2025

Since the text format makes it difficult to read, we have attached a Word document titled "response to reviewers".

Please refer to this Word file.

---

## [Decision Letter · Decision Letter 1]

Medical costs for patients with rheumatoid arthritis who have comorbid diabetes mellitus

PONE-D-24-60159R1

Dear Dr. Tanaka,

We’re pleased to inform you that your manuscript has been judged scientifically suitable for publication and will be formally accepted for publication once it meets all outstanding technical requirements.

Kind regards,

Luca Navarini

Academic Editor

PLOS ONE

Reviewers' comments:

Reviewer's Responses to Questions

**Comments to the Author**

Reviewer #2: All comments have been addressed

2. Is the manuscript technically sound, and do the data support the conclusions?

Reviewer #2: Yes

3. Has the statistical analysis been performed appropriately and rigorously?

Reviewer #2: Yes

4. Have the authors made all data underlying the findings in their manuscript fully available?

Reviewer #2: No

5. Is the manuscript presented in an intelligible fashion and written in standard English?

Reviewer #2: Yes

Reviewer #2: (No Response)

**Do you want your identity to be public for this peer review?** For information about this choice, including consent withdrawal, please see our Privacy Policy

Reviewer #2: **Yes: ** Antonio Ciancio, MD
